# The Effect of Vinegar Supplementation on High-Intensity Cycling Performance within Recreationally Trained Individuals

**DOI:** 10.3390/medicina56090429

**Published:** 2020-08-27

**Authors:** Tyler M. Farney, Robert J. Kowalsky, Dassy A. Salazar, Alyssa N. Fick, Arnold G. Nelson, Christopher M. Hearon

**Affiliations:** 1Human Performance Laboratory, Department of Health & Kinesiology, Texas A&M University-Kingsville, Kingsville, TX 78363, USA; robert.kowalskyjr@tamuk.edu (R.J.K.); dasalazar13@gmail.com (D.A.S.); alyssa.fick@students.tamuk.edu (A.N.F.); christopher.hearon@tamuk.edu (C.M.H.); 2Exercise Biochemistry Laboratory, School of Kinesiology, Louisiana State University, Baton Rouge, LA 70803, USA; anelso@lsu.edu

**Keywords:** acetic acid, cycling, fatigue, glycogen

## Abstract

*Background and objectives*: To investigate the effects of vinegar ingestion upon high-intensity cycle performance in recreationally trained individuals. *Materials and methods*: Twenty-two participants consumed one of the following in a randomized order on four separate visits: (1) 29 mL of vinegar along with 451 mL of water, (2) 39 g of sucrose along with 441 mL of water, (3) 29 mL of vinegar and 39 g of sucrose along with 412 mL of water, or (4) 480 mL of water alone. For each of the experimental testing sessions, all participants completed in order: (1) high-intensity cycle test 1, (2) fatiguing cycle test, (3) high-intensity cycle test 2, (4) supplement consumption, (5) 90 min rest period, and (6) high-intensity cycle test 3. Total time to exhaustion (TTE) and average heart rate (HR) for each set of sprints was used in analysis. *Results*: There was no supplement by time interaction or significant main treatment effect observed (*p* > 0.05) for either TTE or HR. However, there was a main time effect observed, with TTE (*p* = 0.0001) being lower for cycle test 2 than both cycle test 1 and cycle test 3, and cycle test 3 being lower than cycle test 1. HR (*p* = 0.0001) was lower for cycle test 3 than both cycle test 1 and cycle test 2, but HR for cycle test 1 did not differ significantly from HR for cycle test 2. *Conclusions*: The addition of vinegar or sucrose alone, or in combination, was ineffective in improving cycle sprinting TTE when performing three cycle tests.

## 1. Introduction

The ability to maintain muscular force/power production during sport/exercise is at the utmost importance, with any acute impairment in force production or power generation being the primary definition of fatigue [1]. The mechanisms that lead to a loss of force/power are a multi-faceted process. Thus, narrowing down to one primary cause is futile. Therefore, the mechanisms of fatigue should be investigated in a holistic manner. Keeping true to this statement, one important aspect to performance is substrate utilization, particularly glycogen availability. For over two decades, it has been known that the depletion of muscle glycogen leads to a decrease in the ability to produce force or generate power [2]. Subsequently, the ability to rapidly replenish glycogen levels in either the muscle or liver becomes extremely important to maintain both central and peripheral nervous system processes, along with substrate utilization during exercise. Failure to maintain proper glycogen availability will induce difficulty to meet the energetic requirements of training along with during competition [3,4].

As exercise intensity exceeds ~65% VO_2peak_, glycogen becomes the predominate substrate for ATP resynthesis, with the relative use of substrate being dependent upon training status, exercise intensity [5], and exercise duration [6]. In order to increase exercise capacity, it is vital to spare glycogen because as stores of glycogen drop below a certain level, power production falls despite sufficient levels of other fuel sources [7]. Therefore, the goal should be to increase fat oxidation for ATP resynthesis during exercise of moderate intensities and prolonged durations. To help with this, AMP-activated protein kinase (AMPK) plays a key regulatory role on fatty acid metabolism in skeletal muscle during both acute and prolonged exercise [8]. As ATP concentrations decrease and AMP concentrations increase, AMPK becomes activated, as of which this activation has been shown to stimulate an increase in muscle glucose transport and fatty acid oxidation [9,10].

Acetic acid is a major organic acid found in vinegar, and it has been shown to aid in glycogen replenishment within rodents along with the promotion of fatty acid oxidation [11]. The exact mechanisms are unclear. However, acetic acid is believed to help, with increased lipid metabolism and glycogen replenishment through the AMPK signaling pathway [12]. Within the Krebs Cycle of both liver and skeletal muscle, acetic acid is metabolized by way of acetyl-CoA into citrate [13]. The formation of citrate has been found in vitro to inhibit the enzymatic activities of phosphofructokinase type 1 (PFK-1) and type 2 (PFK-2) [14]. Fushimi et al. hypothesized that with the inhibition of glycolysis via the increase in citrate accumulation, acetic acid may increase glycogenesis by increasing the influx of glucose-6 phosphate (G-6-P) into the glycogen synthesis pathway [15]. To highlight this point, Waller and colleagues found that acetate consumption helps to increase muscle glycogen resythensis within horses during a 4 h recovery period [16]. Nakao et al. compared rat gastrocnemius, soleus, and liver glycogen levels following swimming and the ingestion of water, glucose, acetic acid, citric acid, glucose with acetic acid, and glucose with citric acid [17]. It was reported that acetic acid with glucose facilitated liver glycogen restoration during recovery. Additionally, Fushimi et al. reported that feeding rodents glucose with acetic acid accelerated glycogen repletion in skeletal muscle faster than glucose alone [18].

The few published human investigations into acetic acid supplementation are within the arena of athletic training and used pickle juice to determine the impact on performance, particularly in reducing muscle cramps [19,20]. For instance, Peikert et al. reported no significant impact in time to exhaustion, rectal temperature, changes in plasma volume, and sweat volume following the consumption of 2 mL/kg of pickle juice, hypertonic saline, or deionized water [19]. Additionally, Miller et al. reported that consuming 1 mL/kg of pickle juice or deionized water did not exacerbate exercise-induced hypertonicity or cause hyperkalemia [20]. It must be noted that the studies involving the consumption of acetic acid via pickle juice were more focused on the sodium content rather than on the vinegar content. Within the two listed investigations, participants consumed between approximately 80 and 160 mL of pickle juice. Thus, the actual amount of acetic acid within pickle juice was minimal. Therefore, consuming a solution with a greater amount of vinegar might be necessary in order to impact exercise performance within humans.

To our knowledge, no human investigations have been completed on repeated high-intensity exercise performance with the combined supplementation of vinegar and sucrose. As discussed above, a large portion of investigations using vinegar have been completed using animals. Within human investigations, Johnston and colleagues have reported that acetic acid helps to influence hepatic function and metabolic pathways in individuals with Type 2 Diabetes [21,22]. Because of the positive results reported from both animal and clinical investigations, it was warranted to examine if vinegar had a positive impact within apparently healthy humans following high-intensity exercise. Therefore, the purpose of this investigation was to examine the effects of vinegar and sucrose upon repeated high-intensity cycle sprint performance in recreationally trained individuals. Our hypothesis was that supplementing with vinegar and sucrose together would reduce fatigue by improving total time to exhaustion (TTE) via a cycle ergometer.

## 2. Materials and Methods

### 2.1. Study Participants

Twenty-two (males = 15; females = 7) (age: 24.5 ± 5.9 years; stature: 1.69 ± 0.609 m; and mass: 74.8 ± 12.4 kg; all mean ± SD) recreationally trained adults participated in this study. Twenty-four volunteers were recruited. However, two subjects were removed after data collection as outliers based on Chauvenet’s criterion to yield the final sample size (*n* = 22). All 22 participants completed all aspects of this study. Qualification for this study included active involvement in a structured exercise training program (both aerobic and anaerobic) for the previous six months. All participants had to be non-smokers and free from any knee/hip/back injuries. At the start of testing, no participant had any recent surgeries relating to the knee/hip/back. During the first visit, the familiarization session, an explanation of all experimental procedures was provided to all participants. Before study commencement, all participants gave written informed consent, which was approved by the local institutional review board. Before inclusion in this study, all participants completed a Physical Activity Readiness Questionnaire (PAR-Q). Only those who answered “no” to all PAR-Q questions were included. Lastly, prior to beginning the experimental testing sessions, all participants were fully familiarized with the laboratory exercise testing procedures.

### 2.2. Experimental Design

The purpose of this investigation was to determine whether the addition of vinegar and sucrose would help with recovery when performing repeat high-intensity cycle tests to exhaustion. The total amount of acetic acid within vinegar is approximately 5% of total solution, and is considered a safe consumption for humans. All participants reported to the lab for five visits, with the first visit being the familiarization session and the subsequent four visits being experimental testing sessions. A single-blind balanced cross-over design was used with each participant consuming one of the four treatments in a randomized order: (1) 29 mL of a commercially available vinegar along with 412 mL of water and 39 g of sucrose (table sugar), (2) 29 mL of vinegar along with 451 mL of water, (3) 39 g of sucrose along with 441 mL of water, or (4) 480 mL of water alone. The total volume of consumption for all treatments equaled approximately two cups of fluid. The treatments with sucrose and vinegar still equated to approximately two cups of fluid, yet the total solution comprised of approximately 8% sucrose and 5% vinegar. The usage of sucrose was based upon the cost and availability. The total volume of solution was chosen to ensure proper gastric emptying and reduce any distress. For each of the experimental testing sessions, all participants completed in order: (1) high-intensity cycle test 1, (2) fatiguing cycle test, (3) high-intensity cycle test 2, (4) supplement consumption, (5) 90 min rest period, and (6) high-intensity cycle test 3. All testing sessions were conducted on separate days with a one-week “wash-out” period. To maintain dietary consistency, participants were asked on the first day of the experimental testing session to write down what they consumed the day before along with the day of leading up to testing. Each participant received a copy of their consumption and were asked to maintain the same diet throughout the duration of this study.

### 2.3. Maximum Power Test

Following the explanation of all aspects of this study and providing written consent to participate, all participants performed a maximal power test on a cycle ergometer. To begin, participants completed a 5 min warm up on a rate independent cycle ergometer (Lode Corival, Groningen, The Netherlands) with low resistance of 50 watts at a self-selected rate. Following the warm up, participants began pedaling at or above a cadence of 100 rpms with resistance of 100 watts increasing by 25 watts every minute. The increase of 25 watts per minute continued until participants were unable to maintain a cadence of at least 100 rpms for 10 consecutive seconds or were unable to complete a full minute. The wattage for the final completed entire minute determined each participants’ maximal power output, and was used for all experimental sessions. Heart rate (HR) was monitored throughout all tests via a Polar heart rate monitor (Polar Electro, Kempele, Finland).

### 2.4. High-Intensity Cycle Test

The experimental testing sessions consisted of each participant performing a 5 min warm up as performed during the familiarization trial. Following the warm up, participants began cycling for 1 min at 150% of their recorded maximal power output with a cadence above 100 rpms. Participants were free to pedal at any rate above 100 rpms, with the only determinant being to stay above 100 rpms. If the full minute was completed, the participants were given a 2 min rest. Following the rest, participants began cycling again for another minute at 150% of their maximal power output. This continued until participants were unable to either complete a full minute or maintain a cadence of at least 100 rpms for 10 continuous seconds. Heart rate was monitored and recorded following each minute completed. The TTE (sec) and average HR per high-intensity cycle test were used in data analysis. The high-intensity cycle test was completed both before and after a fatiguing cycle test (cycle test 1, cycle test 2), and again following the 90 min post-beverage consumption (cycle test 3). A total of 3 high-intensity cycle tests were completed for all experimental testing sessions.

### 2.5. Fatiguing Cycle Test

The fatiguing cycle test aimed to deplete leg muscle glycogen as much as possible, and began immediately following the high-intensity cycle test 1. This protocol was based off Nelson et al. who investigated muscle glycogen supercompensation following a depletion task [23]. In order to deplete the participants as much as possible, participants cycled on the same cycle ergometer for 30 min at a resistance set at 50% of each participants’ maximal power output. Cadence for the entire 30 min was set at or above 70 rpms. Participants were free to pedal at any rate above 70 rpms, with the only determinant being to stay above 70 rpms. Heart rate was monitored throughout the entire 30 min to ensure participant safety.

### 2.6. Supplementation

Recovery beverages were consumed following the high-intensity cycle test 2 in a single-blind balanced cross-over design. Instructions were to consume the beverage within 20 min of being administered. All participants consumed all four beverages in a randomized order. The beverages used included: (1) 29 mL of commercially available vinegar (Bragg Apple Cider Vinegar, Santa Barbara, CA, USA) mixed in 451 mL of water (VIN), (2) 39 g of sucrose (N’Joy Pure Sugar, New York, NY, USA) mixed in 441 mL of water (SUG), (3) 29 mL of vinegar and 39 g of sucrose mixed in 412 mL of water (CBO), or (4) 480 mL of water alone (H2O).

### 2.7. Statistical Analysis

Two-way (supplement by time) ANOVA with repeated measures was used to analyze for differences between supplement trials (VIN, SUG, CBO, and H2O) across time (high-intensity cycle tests 1–3) for TTE and HR. If needed, appropriate post-hoc tests were used to make pairwise comparisons for specific differences across the experimental sessions and/or time points. The experiment-wise error rate (α = 0.05) was maintained throughout all post-hoc tests for specific differences. For effect size determination, a generalized eta squared following the equations of Bakeman was used [24]. The data are presented as mean ± standard error of the measurement (SEM).

## 3. Results

There was no significant main effect for supplement observed (*p* > 0.05) for either TTE (η^2^ = 0.06) (Table 1) or HR (η^2^ = 0.09) (Table 1). However, there was a significant main effect across time for both TTE (*p* = 0.0001; η^2^ = 0.70) and HR (*p* = 0.001; η^2^ = 0.70). When pooled across supplements, TTE was lower for high-intensity cycle test 2 than both high-intensity cycle test 1 and high-intensity cycle test 3, and high-intensity cycle test 3 was lower than high-intensity cycle test 1 (Table 2). When pooled across supplements, HR was lower for high-intensity cycle test 3 than both high-intensity cycle test 1 and high-intensity cycle test 2, but HR for high-intensity cycle test 1 did not differ significantly from HR for high-intensity cycle test 2 (Table 2). There was no significant supplement by time interaction for either TTE (*p* > 0.05; η^2^ = 0.05) (Figure 1) or HR (*p* > 0.05; η^2^ = 0.05) (Figure 2).

## 4. Discussion

There are relatively few human-based studies investigating the combined effects of vinegar and sucrose. To the best of our knowledge, the current investigation was the first to investigate both supplements on repeated high-intensity cycling performance in human participants. The main findings from this investigation were that neither vinegar combined with sucrose nor vinegar alone had an effect on repeat high-intensity cycling performance within recreationally trained men and women. Both TTE and HR were not different across the four treatments. These findings are contrary to our hypothesis of an improved cycling performance with a reduced rate of fatigue from supplementing with vinegar and sucrose together which we had based upon previous animal research that had used vinegar to help glycogen replenishment. Some limitations that possibly prevented the finding of any statistical significance include the training status of participants, the time of supplementation absorption, and the lack of confirmation of glycogen depletion. Despite not reaching a statistically significant difference with TTE, the VIN group did have a noticeably longer total time compared to the other three treatments. Because of this longer total TTE, vinegar should continue to be investigated under the circumstances that minimize the aforementioned proposed limitations within this investigation.

Within the Peikert investigation, participants started by running at 50% of their age-predicted maximal heart rate for 30 min, followed by treadmill speed increasing 10% every 10 min until 90% of maximal heart rate was reached [19]. If participants were able to run for 10 min at 90% of their maximal heart rate, then intensity was increased to 95% of their maximal heart rate. On average, participants ran for approximately 77 min with rectal temperature increasing to just below 39 °C. It was believed that both the pickle juice and hypertonic saline would delay TTE and lower rectal temperature due to the sodium in both treatments along with the pickle. However, all the tested variables were unaltered following exercise. The speculation was that the consumed volumes of either treatment were insufficient to increase extracellular sodium concentrations, and subsequently alter osmotic fluid pressure. It has been reported that with plasma volume expansion, time to exhaustion is extended out by 6 min when exercising at 87% to 91% of peak oxygen consumption [25], and 20.8 min when intensity was set at 70% maximal oxygen consumption [26]. The same situation could be the reason behind the lack of a significant difference in our TTE. It is possible that despite providing the participants with approximately two cups of fluid that included 29 mL of vinegar, the total amount of fluid was insufficient to increasing osmotic pressure and alter performance. On the other hand, our participants consumed no other fluids during the 90 min rest period, which should have provide sufficient time to allow for proper gastric emptying even when consuming a greater volume of vinegar.

One of the main proposed benefits of acetic acid is its ability to load glycogen at a faster rate when consumed along with glucose [11,18]. Within the animal investigations that found acetic acid to promote a greater glycogen replenishment, rodents exercised from 80 min [11], to 120 min [17,18]. The Nakao investigation had rodents swim for 120 min until exhaustion followed by ingestion of different treatments with a total volume of 2 mL, two of which being 0.6 g of glucose with either 0.048 g of acetic acid or 0.1 g of citric acid. The authors reported a significant reduction in glycogen levels of the liver, gastrocnemius, and soleus following the exhaustive exercise, however, the glycogen levels in both the liver and gastrocnemius were significantly higher with the combination of glucose/acetic acid and glucose/citric acid [17]. Only the combination of glucose/citric acid raised soleus muscle glycogen levels significantly. Fushimi et al. followed a similar protocol with rodents exercising for 120 min, and reported that feeding glucose with acetic acid helped to accelerate the repletion of glycogen levels greater than glucose alone [18]. Taken together, these results help to support that glycogen is replenished to a greater extent with the addition of acetic acid when glycogen levels actually get depleted to a significant level. Thus, one possible explanation for the non-significant change in TTE within the current investigation was that glycogen was not sufficiently depleted to a point where additional glycogen was needed to induce a change in work time. There was a significant reduction in TTE from high-intensity cycle test 1 to high-intensity cycle test 2, which followed the fatiguing cycle test. However, despite fatigue occurring from high-intensity cycle test 1 to high-intensity cycle test 2, the overall stimulus from the combined cycling tests and steady state bout may not have been great enough to lower glycogen levels to where acetic acid could help with performance.

Finally, substrate use during exercise is influenced by different factors such as conditioning level, workload amounts and work duration. As mentioned above, Waller and colleagues found that acetic acid sped glycogen resynthesis in racehorses [16]. With respect to condition levels, one can assume that trained racehorses are more trained than recreationally active individuals used in this study. Additionally, the trained state among the horses was more similar than that among our subjects, as evidenced by the wide range of cycling times to exhaustion among our subjects. Furthermore, the work performed by a horse is much higher than that of a human doing weight supported cycling. Thus, the use of recreationally active individuals placed a limitation on establishing a benefit from acetic acid supplementation. Therefore, future investigations should incorporate a more sport-specific approach to the methodology by using trained athletes as participants.

## 5. Conclusions

The current investigation was the first to investigate the impact of acetic acid via vinegar supplementation following three separate high-intensity cycling tests. Despite not reaching statistical significance, supplementing with vinegar allowed for a greater TTE on a cycle ergometer. Due to this, coupled with the positive findings from animal investigations, acetic acid via vinegar consumption should continue to be investigated. Vinegar supplementation may be beneficial if studied within a study design that is more in line with an actual sporting event along with a more precise dosage. Specifically, this would include studying how highly trained individuals respond to vinegar consumption based upon body mass in a glycogen-depleting events that consists of multiple periods of high-intensity work separated by rest periods of several minutes. Acetic acid has been reported to improve performance due to its ability to increase lipid metabolism and glycogen replenishment. Future investigations should investigate vinegar supplementation among highly trained athletes within a real world environment, such as a competition where intense work bouts are separated by rest periods. A good example of this competition would be an ice hockey game, where players are performing multiple repeat bouts of intense work with rest periods. Nonetheless, protocols need to ensure that exercise stimulus is great enough to more fully deplete glycogen stores along with a proper dosage to ensure cellular uptake. Combining the proper stimulus with proper dosage may allow for acetic acid to fully help with replenishment, thus, increasing exercise performance.

## Figures and Tables

**Figure 1 medicina-56-00429-f001:**
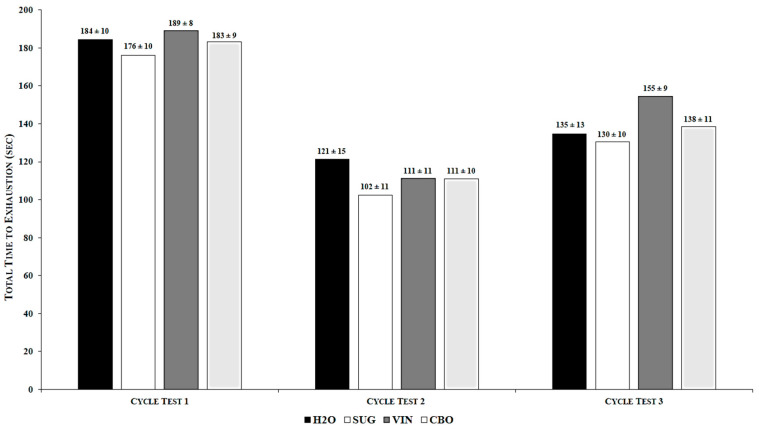
Supplement by Time Interaction for Total Time to Exhaustion. Values are the mean ± SEM; *p* = 0.37. H2O: water; SUG: sugar; VIN: vinegar; CBO: combination.

**Figure 2 medicina-56-00429-f002:**
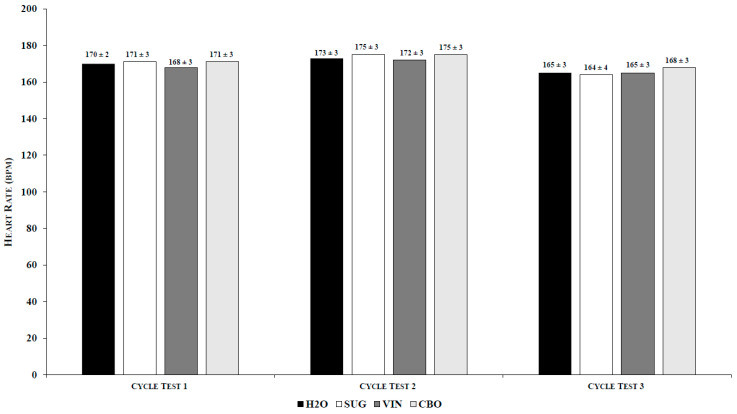
Supplement by Time Interaction for Heart Rate. Values are the mean ± SEM; *p* = 0.34. H2O: water; SUG: sugar; VIN: vinegar; CBO: combination.

**Table 1 medicina-56-00429-t001:** Main Effect for Supplement.

Variable	Water (H2O)	Sugar (SUG)	Vinegar (VIN)	SUG + VIN Combination (CBO)	*p*-Value
Total Time to Exhaustion (secs)	147 ± 8	136 ± 7	152 ± 7	144 ± 7	0.25
Heart Rate (bpm)	169 ± 2	170 ± 2	168 ± 2	171 ± 2	0.11

Values are the mean ± SEM.

**Table 2 medicina-56-00429-t002:** Main Effect for Time.

Variable	Cycle Test 1	Cycle Test 2	Cycle Test 3	*p*-Value
Total Time to Exhaustion (secs)	183 ± 5 ^a^	111 ± 6 ^a^	139 ± 5 ^a^	0.0001
Heart Rate (bpm)	175 ± 2 ^b^	174 ± 1 ^c^	165 ± 2 ^b,c^	0.0001

Values are the mean ± SEM. Superscript ^a^ denotes cycle sprints that differed significantly for TTE. Superscript ^b,c^ denotes cycle sprints that differed significantly for HR.

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
