# Peer review of "The Effect of Vinegar Supplementation on High-Intensity Cycling Performance within Recreationally Trained Individuals"

_medicina, 2020, doi:10.3390/medicina56090429_

Round 1
Reviewer 1 Report
per response on the agreement page, the authors seem to have addressed the noted issues adequately to be appropriate for publication
Reviewer 2 Report
No more comments.
This manuscript is a resubmission of an earlier submission. The following is a list of the peer review reports and author responses from that submission.
Round 1
Reviewer 1 Report
In this manuscript, the authors share their findings related to vinegar supplementation on cycling performance in recreationally trained males and females. Although the population may have led to some nonsignificant findings, the novel approach of vinegar supplementation in human performance is quite interesting. Overall the paper is well written, and the findings provide a good base for human performance researchers to build upon. Specific comments with line numbers (based on the submitted PDF) follow:
Title: since your study is novel to human populations, suggest including "recreationally trained males and females" into your title. Waller and Nakao, papers you cite multiple times, mention horses and rats in their titles, respectively. Perhaps the addition of "humans" or the prior suggestion helps differentiate your work...
Line 21: "Methods and methods"...?
Abstract: Suggest mentioning the post hoc analyses conducted to assess pairwise differences
Line 51: The use of forward slash for "force/power" is one thing, but to use it repeatedly at "produce force/generate power" and "and/or" (line 56) seems a little sloppy. Just write out the sentences fully.
Line 68: Extra space seems to exist at "transport and"
Line 69: Acetic acid is "A" major organic...
Line 71: You mention several rodent investigations within this paragraph to speak about the potential benefits to human performance. Although my work with rodent research is non-existent, citations 11 and 17 appears to be more endurance-based activity. Does the methodology presented in those papers match yours and properly reflect the substrate utilization you would see in your human trials?
Line 80: "helps" rather than "help"?
Line 99: you mention consumption of 80-160 mL of pickle juice. Your participants consumed >400mL, with a certain amount of vinegar etc. What is the concentration of acetic acid in pickle juice, and how might this connect to your dosage?
Line 115: I believe this is the first mention of "time to exhaustion" in the body of your paper. You operationally define "TTE" at line 178, however. Furthermore, despite defining TTE, you continue to use "time to exhaustion" throughout the paper. Suggest defining TTE at first mention and using TTE thereafter. Examine throughout the document for this inconsistency.
Line 119: You mentioned some stipulations for inclusion criteria, but what was the specific definition of "recreationally trained". And since you state it as a potential limitation, what was the rationale for this population initially?
Line 121: You reference Chauvenet's criterion as a justification to drop two participants from your study. Chauvenet's criterion seems to imply moreso that the data should only be omitted if it is spurious in nature. Was the data from either of these two participants plausible, despite either having lower or higher performance than the remaining 22 participants? It is hard to believe that all four sessions for each of the two participants was spurious. If so, was there some sort of equipment calibration issue etc. unique to the four/five visits of each of the participants? If not, simply having skewed performance is not justification enough for omitting their data.
Participants: Was diet or hydration level controlled for across the repeated visits?
Line 158: Was the 5-minute warm-up at a self-selected pace?
Line 180: "was completed both before and after a fatiguing cycle test" -- it took me a minute to figure out that "fatiguing cycle" test and "30 minute continuous cycling" was the same thing. Prior to this line, you refer to it as "30 min..." and after this line, you seem to refer to it as "fatiguing test". Consider revising so it is referred consistently throughout the manuscript.
Line 186: Possessive needs to be added -- "Nelson's and colleagues'"
Line 192: Arbitrary space jump to next line
Line 193: Was HR monitored via chest strap? Mention instrumentation.
Line 200: Is there a variation of the acetic acid or potency of the vinegar depending on the type of vinegar? i.e. apple cider vinegar vs distilled white vs rice wine vinegar? If so, why was apple cider vinegar selected?
Figure 1 and 2: Since the supplement was not introduced until after "cycle test 2" what is the rationale for included all conditions for cycle test 1 and 2? I yield that it does help show the increase of TTE for VIN, but it highlights a potential issue if you did not control for diet and hydration (see earlier comment)
Line 253: "Reduced rate" rather than "reduce rate"
Line 259: See earlier comment regarding TTE/time to exhaustion
Line 277: Inconsistent citation after "consumption"
Line 286: You seem to use "acetic acid", "acetic vinegar", and "vinegar" interchangeably throughout the paper. For consistency sake, I would suggest using "acetic vinegar" or "vinegar" when speaking about your results as you differentiate it enough from "acetic acid" at Line 69.
Line 300: Did you mean to say "the addition of glucose to acetic acid"? The sentences leading up to this seem to highlight the combination of glucose and acetic acid in the other studies.
Line 314: The sentence comparing training status of racehorses and recreational adults leads to the question regarding dosage. Was the rodent dosage in Fushimi, Nakao, etc consistent with horse dosage in Waller consistent with your dosage? Given considerable difference in mass, metabolism, muscle fiber type, etc. across the three "participant" groups, I am wondering if that may play a role in the inconsistency of findings across the animal/human studies
Reviewer 2 Report
address as to the use of sucrose vs glucose as the added substance in the the study; perhaps strengthen if known levels of key subsances prior to intense exercise.
The primary (main) question addressed in this paper seemed to be clearly stated and considering the current state of athletes (recreational as well as elite) with regard to utilizing supplementation such as pickle juice and vinegar in order to advance in their sport, the question is certainly a novel one and of both interest and relevance. In agreement with the comments in the reported introduction, this is an original paper / topic with regard to moving research from the animal/test tube environment to humans and in their competitive environments. In comparison to other reported research in this area, it is likely to become one that is adding information to the initial study of the topic.
I found the paper to mostly be clearly stated and presenting facts in a relevant and understandable manner. As noted previously, clarification on some of the terms and consistency with the use of those terms - TTE vs failure. (noted in the abstract in particular; line37 p 1; A primary concern was the use of sucrose without explanation as to why it was selected.
The information would be enhanced with some indications as the glycogen storage levels of the subjects before the trials as well as levels post trial.
Conclusions seemed consistent with the evidence that was presented and though they did not result in a positive significant finding, the information is of great value to the knowledge and laid a foundation for future and perhaps more controlled research to be completed. For future clarity, it likely would be helpful for the question to be more precise since the results as recorded seemed more specific than the broader research question.
This paper would be presenting more persuasive/compelling information if additional clarity was noted as to the actual metabolic actions that were anticipated would be noted more extensively in order to support the use of vinegar and sucrose rather than the earlier work that addressed acetic acid/citric acid/ and glucose.
Reviewer 3 Report
The article examined relatively healthy active individuals. Two high intensity cycling bouts followed by a 30 minute continuos cycling bout was used to deplete glycogen, followed by four randomized treatment supplements. The main goal of the study was to examine the effects of 4 treatments (sucrose and water; water alone; vinegar and water; and the combination of vinegar, sucrose, and water). The Supplement was given after high-intensity cycling test 1, 30 minutes of continuous cycling, high intensity cycling test 2. Following the consumption of the randomized supplement, participants completed a 3rdhigh intensity cycle test.
This paper provides a glimpse into the effects of vinegar and glycogen replenishment in humans when this technique has been effective in rodents and horses. Although there was no significance found between the four treatments, minimizing the limitations and testing in highly trained and sport specific individuals could elicit a positive outcome.
Strengths: This was a novel study utilizing a vinegar supplement in humans as opposed to animal studies used previously. There appears to be a strong power with the number of participants that completed all four trials.
Weaknesses: More attention to detail when controlling the participants. A main concern for the authors was not knowing if the exercise bouts were enough to deplete glycogen stores. More citations with the exercise bout eliciting glycogen depletion in humans would have provided that information. Also checking for glycogen depletion with multiple other devices such as blood glucose, VO2, or ultrasound.
Minor grammatical errors in the intro
Line 84-85: The statement is supplementing acetic acid and glucose for liver glycogen post exercise. What about during exercise? Glycogen synthesis is a slow process. Please explain the importance of not only resynthesizing glycogen during exercise, but also limiting glycogen depletion during exercise.
Materials and Methods: Did you test any baseline glucose on the participants? I believe that information needs to be mentioned because any elevated fasted glucose or impaired glucose tolerance will play a major role in both glycogenesis and glycogenolysis. Please explain reasoning.
Did participants test at the same time of day each time? Did you monitor blood or plasma glucose concentration at any time during the testing procedures? Some participants may have arrived fasted and hypoglycemic, or immediately following a meal and hyperglycemic.
Was VO2collected at any point during the study to monitor respiratory exchange ratio and substrate utilization?
Results: I cannot see a legend for the bar graphs. Did you only pool the cycle tests for time to exhaustion? You should isolate the 3rdhigh intensity cycle test to determine the effect of each supplement.
Discussion:
Line 260: Add the percent differences to express a better understanding even though the VIN group was still not significantly different
Line 271: The cited study also examined plasma volume and sweat volume. Should the authors have given supplement doses based on body mass? Please explain the differences in further detail.